# Integrin β4 promotes DNA damage-related drug resistance in triple-negative breast cancer via TNFAIP2/IQGAP1/RAC1

Huan Fang[1,2†], Wenlong Ren[1,3†], Qiuxia Cui[1,4,5†], Huichun Liang[1†], Chuanyu Yang[1], Wenjing Liu[1], Xinye Wang[1], Xue Liu[6], Yujie Shi[7]*, Jing Feng[6,8,9]*, Ceshi Chen[1,10,11]*

[1]Kunming Institute of Zoology, Chinese Academy of Sciences, Kunming, Yunnan, China; [2]Kunming College of Life Sciences, University of Chinese Academy of Sciences, Kunming, Yunnan, China; [3]School of Life Science, University of Science & Technology of China, Hefei, China; [4]Affiliated Hospital of Guangdong Medical University, Guangdong, China; [5]Department of Breast Surgical Oncology, National Cancer Center/National Clinical Research Center for Cancer/Cancer Hospital & Shenzhen Hospital, Chinese Academy of Medical Sciences and Peking Union Medical College, Shenzhen, China; [6]Shanghai University of Medicine & Health Sciences Affiliated Sixth People's Hospital South Campus, Shanghai, China; [7]Department of Pathology, Henan Provincial People's Hospital, Zhengzhou University, Zhengzhou, China; [8]The Second Affiliated Hospital of the Chinese University of Hong Kong (Shenzhen), Shenzhen, China; [9]School of Laboratory Medicine and Biotechnology, Southern Medical University, Guangdong Province, Guangzhou, China; [10]Academy of Biomedical Engineering, Kunming Medical University, Kunming, China; [11]The Third Affiliated Hospital, Kunming Medical University, Kunming, China

**\*For correspondence:**
iyujie523@126.com (YS);
fengjing71921@163.com (JF);
chenc@mail.kiz.ac.cn (CC)

[†]These authors contributed equally to this work

**Abstract** Anti-tumor drug resistance is a challenge for human triple-negative breast cancer (TNBC) treatment. Our previous work demonstrated that TNFAIP2 activates RAC1 to promote TNBC cell proliferation and migration. However, the mechanism by which TNFAIP2 activates RAC1 is unknown. In this study, we found that TNFAIP2 interacts with IQGAP1 and Integrin β4. Integrin β4 activates RAC1 through TNFAIP2 and IQGAP1 and confers DNA damage-related drug resistance in TNBC. These results indicate that the Integrin β4/TNFAIP2/IQGAP1/RAC1 axis provides potential therapeutic targets to overcome DNA damage-related drug resistance in TNBC.

## eLife assessment

This study presents a rather **valuable** finding that IQGAP1 interacts with TNFAIP2, which activates Rac1 to promote drug resistance in TNBC. The evidence supporting the claims of the authors is quite **solid**. The work will be of interest to scientists working on breast cancer.

## Introduction

Breast cancer is the most commonly diagnosed cancer and the leading cause of cancer death in women (*Bray et al., 2018*). Although the diagnosis and treatment of breast cancer has entered the era of molecular typing, 35% of breast cancers still experience recurrence, metastasis, and treatment failure (*Zhao et al., 2020*). According to the expression of estrogen receptor (ERα), progesterone receptor (PR), and human epidermal growth factor receptor (HER2), breast cancer is divided into ER/

PR-positive, HER2-positive, and triple-negative breast cancer (TNBC) (*Sotiriou and Pusztai, 2009*). For ER/PR- and HER2-positive breast cancer, endocrine therapies such as tamoxifen and anti-HER2 targeted therapy such as trastuzumab have achieved good efficacy. Targeted drugs for TNBC patients with BRCA1/2 mutations include two poly ADP-ribose polymerase (PARP) inhibitors, olaparib and talazoparib. These targeted drugs cannot fully meet the clinical needs of patients with various TNBC subtypes (*Bai et al., 2021*). Currently, DNA damage chemotherapy drugs, including epirubicin and cisplatin, are widely used for TNBC treatment.

TNBC often recurs and metastasizes due to the development of chemoresistance, although it is initially responsive to chemotherapeutic drugs (*Jamdade et al., 2015*). Chemoresistance severely impacts the clinical outcomes of patients. Tumor cells become resistant to chemotherapeutic agents through several mechanisms, such as improving DNA damage repair, changing the intracellular accumulation of drugs, or increasing anti-apoptotic mechanisms (*Hill et al., 2019*). Therefore, characterization of the underlying molecular mechanisms by which resistance occurs will provide opportunities to develop precise therapies to enhance the efficacy of standard chemotherapy regimens (*Longley and Johnston, 2005*; *Rincón et al., 2016*).

TNFAIP2 is abnormally highly expressed in a variety of tumor cells, including TNBC (*Jia et al., 2016*), nasopharyngeal carcinoma (*Chen et al., 2011*), malignant glioma (*Cheng et al., 2015*), uroepithelial carcinoma (*Niwa et al., 2019*), and esophageal squamous cell carcinoma (*Xie and Wang, 2017*), and is associated with poor prognosis. Our previous work (*Jia et al., 2016*; *Jia et al., 2018*) showed that TNFAIP2, as a KLF5 downstream target protein, can interact with RAC1 (*Didsbury et al., 1989*), a member of the Rho small GTP enzyme family, and activate RAC1 to alter the cytoskeleton, thereby inducing filopodia and lamellipodia formation and promoting the adhesion, migration, and invasion of TNBC cells. After activation, RAC1 can activate AKT, PAKs, NADPH oxidase, and other related signaling pathways to promote cell survival, proliferation, adhesion, migration, and invasion (*Rul et al., 2002*). Activation of RAC1 can reduce the therapeutic response to trastuzumab in breast cancer and increase the resistance of TNBC cells to paclitaxel (*Liu et al., 2019*), but the specific mechanism of action is not completely clear.

RAC1 has been shown to play an important role in DNA damage repair. Activated RAC1 can promote the phosphorylation of the DNA damage response-related molecules ATM/ATR, CHK1/2, and H2AX by activating the activity of protein kinases such as ERK1/2, JNK, and p38 (*Yan et al., 2012*; *Wu et al., 2019*), thus improving the DNA damage repair ability and inhibiting tumor cell apoptosis (*Hu et al., 2016*; *Li et al., 2020*; *Hervieu et al., 2020*). RAC1 also promotes aldolase release and activation by changing the cytoskeleton and activates the ERK pathway to increase the pentose phosphate pathway to promote nucleic acid synthesis, providing more raw materials for DNA damage repair (*Li et al., 2021*; *Feng et al., 2010*). At the same time, the interaction of RAC1 and PI3K promoted AKT phosphorylation and glucose uptake (*Higuchi et al., 2008*; *Hu et al., 2018*). Therefore, RAC1 is well established to promote the chemoresistance of breast cancer by promoting DNA damage repair.

Integrin β4 (ITGB4) is a major component of hemidesmosomes and a receptor molecule of laminin. Studies have shown that laminin-5 interacts with ITGB4 to activate RAC1 activity and promote cell migration (*Hamill et al., 2009*) and polarization (*Wu et al., 2018*) by altering the cytoskeleton. Since ITGB4-positive cancer stem cell (CSC)-enriched mesenchymal cells were found to reside in an intermediate epithelial/mesenchymal phenotypic state, ITGB4 can be used to enable stratification of mesenchymal-like TNBC cells (*Bierie et al., 2017*). In addition, the expression of ITGB4 on ALDH$^{high}$ breast cancer and head and neck cancer cells was significantly greater than that on ALDH$^{low}$ cells, proving the effects that ITGB4 targets on both bulk and CSC populations (*Ruan et al., 2020*). Furthermore, ITGB4-overexpressing TNBC cells provided cancer-associated fibroblasts (CAFs) with ITGB4 proteins via exosomes, and ITGB4-overexpressing CAF-conditioned medium promoted the proliferation, epithelial-to-mesenchymal transition, and invasion of breast cancer cells (*Sung et al., 2020*). ITGB4 also promotes breast cancer cell resistance to tamoxifen-induced apoptosis by activating the PI3K/AKT signaling pathway and promotes breast cancer cell resistance to anoikis by activating RAC1 (*Kim et al., 2012*). However, how ITGB4 activates RAC1 is not completely clear.

RAC1 activity is regulated by guanylate exchange factors (GEFs), GTPase activation proteins (GAPs), and guanine separation inhibitors (GDIs) (*Cherfils and Zeghouf, 2013*). GAPs typically provide the necessary catalytic groups for GTP hydrolysis, but not all GAPs function as hydrolases. IQGAP1 lacks

an arginine in the GTPase binding domain and cannot exert the hydrolysis effect of GAPs (*Schmidt, 2012*). IQGAP1 can increase the activity of RAC1 and CDC42 (*Smith et al., 2015*; *Gorisse et al., 2020*).

In this study, we demonstrated that TNFAIP2 interacts with IQGAP1 and ITGB4. ITGB4 promotes TNBC drug resistance via the TNFAIP2/IQGAP1/RAC1 axis by promoting DNA damage repair. Our results suggest that ITGB4 and TNFAIP2 might serve as promising therapeutic targets for TNBC.

## Results

### TNFAIP2 promotes TNBC DNA damage-related drug resistance

To explore the functional significance of TNFAIP2 in TNBC drug resistance, we constructed stable TNFAIP2 overexpression and TNFAIP2 knockdown HCC1806 and HCC1937 cells. As shown in *Figure 1A–E*, overexpression of TNFAIP2 significantly increased cell viability when treated with Epirubicin (EPI) and Talazoparib (BMN). Additionally, knockdown of TNFAIP2 significantly decreased cell viability when treated with EPI and BMN (*Figure 1F–J*). We then examined the effects of TNFAIP2 on DNA damage repair and found that TNFAIP2 promotes DNA damage repair in response to EPI and BMN. TNFAIP2 overexpression decreased the protein expression levels of γH2AX, a marker of DNA damage, and cleaved-PARP, a marker of apoptosis (*Figure 1K*). Additionally, knockdown of TNFAIP2 significantly increased γH2AX and cleaved-PARP protein expression levels in response to EPI and BMN in both cell lines (*Figure 1L*).

The function of TNFAIP2 was further validated by using two other DNA damage drugs, DDP and AZD (*Figure 1—figure supplement 1A–L*). These results suggested that TNFAIP2 enhances TNBC cell drug resistance by promoting DNA damage repair.

### TNFAIP2 confers TNBC drug resistance in vivo

To test whether TNFAIP2 also decreases the sensitivity of TNBC cells to EPI and BMN in vivo, we performed animal experiments in nude mice. HCC1806 cells with stable TNFAIP2 knockdown were orthotopically inoculated into the fat pad of 7-week-old female mice ($n$ = 8 or 12/group). Western blotting (WB) was performed to detect the knockdown effect of TNFAIP2 protein in animal experiments (*Figure 2—figure supplement 2G*). When the tumor mass reached approximately 50 mm³, each group was divided into two subgroups to receive either EPI (2.5 mg/kg, twice a week) or vehicle control for 23 days and either BMN (1 mg/kg, twice a week) or vehicle control for 29 days. We observed that depletion of TNFAIP2 suppressed breast cancer cell growth in vivo. This is consistent with our previous report (*Jia et al., 2016*). More importantly, TNFAIP2 depletion further decreased tumor volume when mice were treated with EPI and BMN (*Figure 2A–F*). Meanwhile, BMN treatment had no effect on the body weight of mice (*Figure 2—figure supplement 1F*). Consistently, EPI and DDP generated similar results but decreased mouse body weight due to their high toxicity (*Figure 2—figure supplement 1D, E*). These results suggest that inhibition of TNFAIP2 expression can overcome HCC1806 breast cancer cell drug resistance in animals.

### TNFAIP2 promotes TNBC drug resistance and DNA damage repair via RAC1

Since chemotherapeutic agents and PRAP inhibitors induce DNA damage directly or indirectly, DNA damage repair ability profoundly affects the sensitivity of cancer cells to these therapies (*Woods and Turchi, 2013*; *Cheung-Ong et al., 2013*). Since TNFAIP2 can activate RAC1, a well-known drug resistance protein, we investigated whether TNFAIP2 induces chemotherapeutic resistance through RAC1. We found that RAC1 knockdown abrogated the effects of TNFAIP2 overexpression-induced drug resistance to EPI and BMN in HCC1806 and HCC1937 cells (*Figure 3A–F*). We also found that γH2AX and cleaved-PARP protein levels were upregulated again in RAC1 knockdown and TNFAIP2-overexpressing HCC1806 and HCC1937 cells in response to EPI and BMN (*Figure 3G–J*). We obtained similar results by using DDP and AZD treatment (*Figure 3—figure supplement 1A–J*). Collectively, these results suggest that TNFAIP2 promotes DNA damage repair and drug resistance via RAC1.

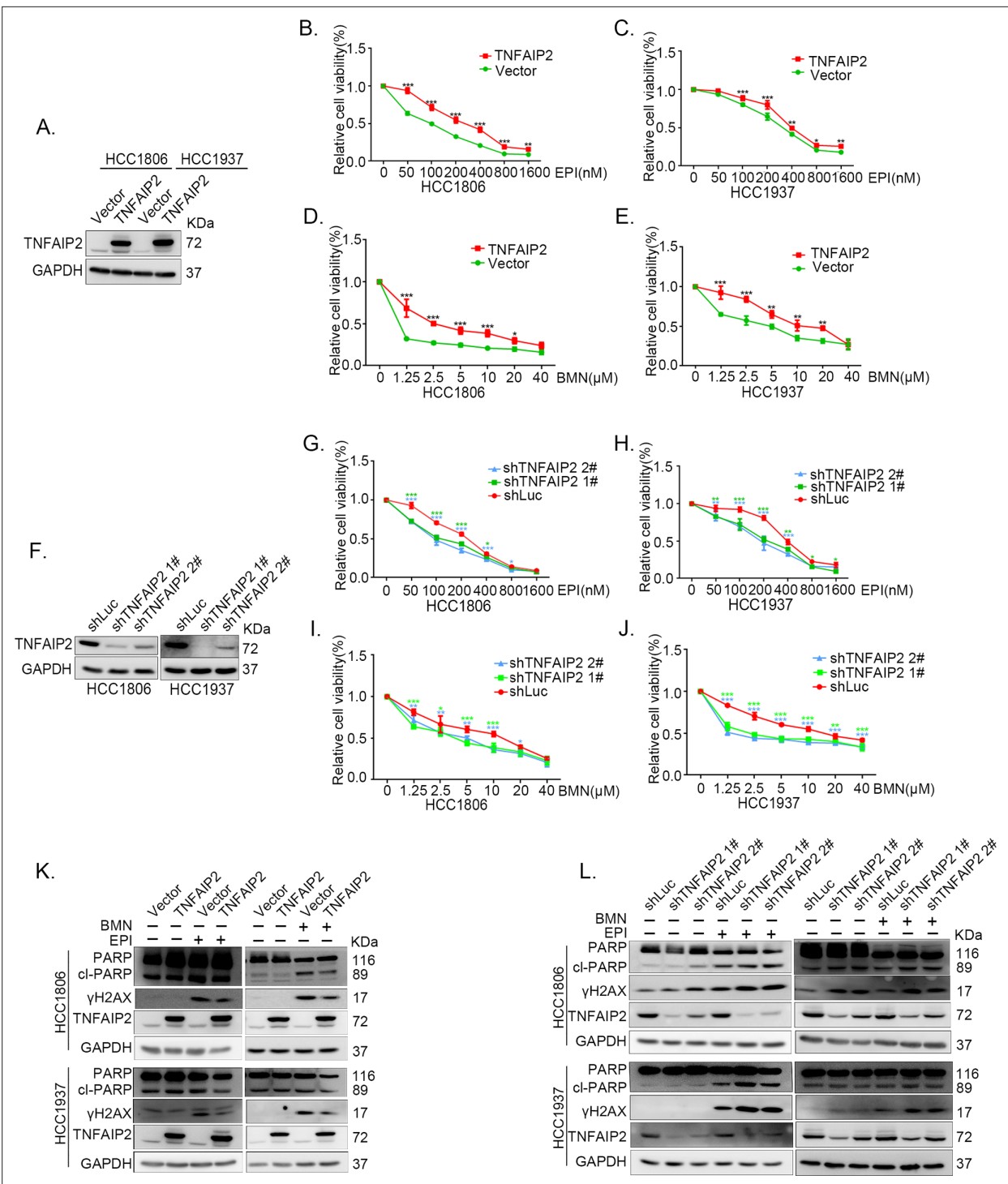

**Figure 1.** TNFAIP2 promotes triple-negative breast cancer (TNBC) DNA damage-related drug resistance. (**A–E**) Stable TNFAIP2 overexpression in HCC1806 and HCC1937 cells significantly increased cell viability in the presence of EPI (0–1.6 μM) or BMN (0–40 μM) treatment for 48 hr, as measured by the SRB assay. Statistical analysis was performed using one-way analysis of variance (ANOVA), n = 3, *p < 0.05, **p < 0.01, ***p < 0.001. TNFAIP2 protein expression was detected by western blotting (WB). (**F–J**) Stable TNFAIP2 knockdown in HCC1806 and HCC1937 cells significantly decreased cell viability in the presence of EPI (0–1.6 μM) or BMN (0–40 μM) treatment for 48 hr, as measured by the SRB assay. Statistical analysis was performed using one-way ANOVA, n = 3, *p < 0.05, **p < 0.01, ***p < 0.001. TNFAIP2 protein expression was detected by WB. (**K**) TNFAIP2 promoted DNA damage repair in the presence of EPI and BMN. HCC1806 and HCC1937 cells stably overexpressing TNFAIP2 were treated with 400 or 800 nM EPI for 48 hr and 10 μM BMN for 24 hr, respectively. TNFAIP2, γH2AX, and PARP protein expression was detected by WB. (**L**) TNFAIP2 knockdown increased DNA damage in

*Figure 1 continued on next page*

*Figure 1 continued*

the presence of EPI and BMN. Stable TNFAIP2 knockdown cells were treated with 400 or 800 nM EPI for 24 or 48 hr and 2.5 µM BMN for 24 hr. TNFAIP2, γH2AX, and PARP protein expression was detected by WB.

The online version of this article includes the following source data and figure supplement(s) for figure 1:

**Source data 1.** Uncropped western blot images for *Figure 1*.

**Figure supplement 1.** TNFAIP2 promotes triple-negative breast cancer (TNBC) DNA damage-related drug resistance.

**Figure supplement 1—source data 1.** Uncropped western blot images for *Figure 1—figure supplement 1*.

## IQGAP1 mediates RAC1 activation by TNFAIP2 and promotes TNBC drug resistance

To characterize the mechanism by which TNFAIP2 activates RAC1, we performed an immunoprecipitation and silver staining (IP-MS) experiment. We found that TNFAIP2 interacts with IQGAP1 (*Figure 4A*). To validate whether TNFAIP2 interacts with IQGAP1, we constructed HCC1806 cells with stable Flag-TNFAIP2 overexpression and collected Flag-tagged TNFAIP2 cell lysates for immunoprecipitation assays using Flag-M2 beads (*Figure 4—figure supplement 1A*).We performed immunoprecipitation using ananti-IQGAP1 antibody and found that endogenous IQGAP1 protein interacted with endogenous TNFAIP2 protein inHCC1806 cells (*Figure 4B*). Next, we mapped the regions of TNFAIP2 and IQGAP1 proteins responsible for the interaction by generating a series of Flag-TNFAIP2 deletion mutants and transfected them into HEK293T cells together with full-length IQGAP1. Then, we performed immunoprecipitation assays using Flag-M2 beads (*Figure 4—figure supplement 1B*). We demonstrated that the N-terminus (1–79 aa) of the TNFAIP2 protein interacted with IQGAP1. To explore the function of IQGAP1 in TNBC drug resistance, we knocked down IQGAP1 in HCC1806 and HCC1937 cells. As shown in *Figure 4C–G*, knockdown of IQGAP1 significantly decreased cell viability in the presence of EPI and BMN in both cell lines. We also examined the effects of IQGAP1 on DNA damage repair and found that IQGAP1 promotes DNA damage repair. IQGAP1 knockdown increased γH2AX and cleaved-PARP protein expression levels when HCC1806 and HCC1937 cells were treated with EPI and BMN (*Figure 4H*). Next, we found that IQGAP1 knockdown abrogated the effects of TNFAIP2 overexpression on resistance to EPI and BMN (*Figure 4I–K*, *Figure 4—figure supplement 1C–E*). We also found that γH2AX and cleaved-PARP protein levels were upregulated in IQGAP1 knockdown and TNFAIP2-overexpressing HCC1806 and HCC1937 cells (*Figure 4L*). In addition, we found that the TNFAIP2 overexpression-induced increase in RAC1 activity was abolished by IQGAP1 knockdown (*Figure 4M*).

## ITGB4 interacts with TNFAIP2 and promotes TNBC drug resistance and DNA damage repair

In addition to IQGAP1, TNFAIP2 may interact with ITGB4 (*Figure 4A*). To validate whether TNFAIP2 interacts with ITGB4, we immunoprecipitated exogenous Flag-tagged TNFAIP2 proteins from HCC1806 cells by using Flag-M2 beads and detected endogenous ITGB4 proteins (*Figure 5A*).To further confirm the protein–protein interaction between endogenous TNFAIP2 and ITGB4 proteins, we collected HCC1806 cell lysates and performed immunoprecipitation using an anti-TNFAIP2/ITGB4 antibody and found that endogenous TNFAIP2/ITGB4 protein interacted with endogenous ITGB4/TNFAIP2 protein (*Figure 5B, C*). We further mapped the regions of TNFAIP2 and ITGB4 proteins responsible for the interaction (*Figure 5—figure supplement 2I*) by generating a series of Flag-TNFAIP2/GST-fused TNFAIP2 deletion mutants and transfected them into HEK293T cells together with full-length GST-fused ITGB4/ITGB4. Then, we performed immunoprecipitation assays using Flag-M2 beads and glutathione beads. As shown in *Figure 5—figure supplement 2J, K*, the N-terminus (218–287 aa) of the TNFAIP2 (TNFAIP2-S-N1-3) protein interacted with ITGB4. To map the domains of ITGB4 that interact with TNFAIP2, we transfected Flag-tagged full-length TNFAIP2 into HEK293T cells with full-length or truncated ITGB4. We found that the C-terminus (710–740 aa) of the ITGB4 protein interacted with TNFAIP2 (*Figure 5—figure supplement 2L, M*). Taken together, these results suggest that TNFAIP2 interacts with ITGB4 and that their interaction is mediated through the N-terminus of TNFAIP2 and the C-terminus of ITGB4.

To explore the function of ITGB4 in TNBC drug resistance, we knocked down ITGB4 in HCC1806 and HCC1937 cells. As shown in *Figure 5D–I*, knockdown of ITGB4 significantly decreased cell viability

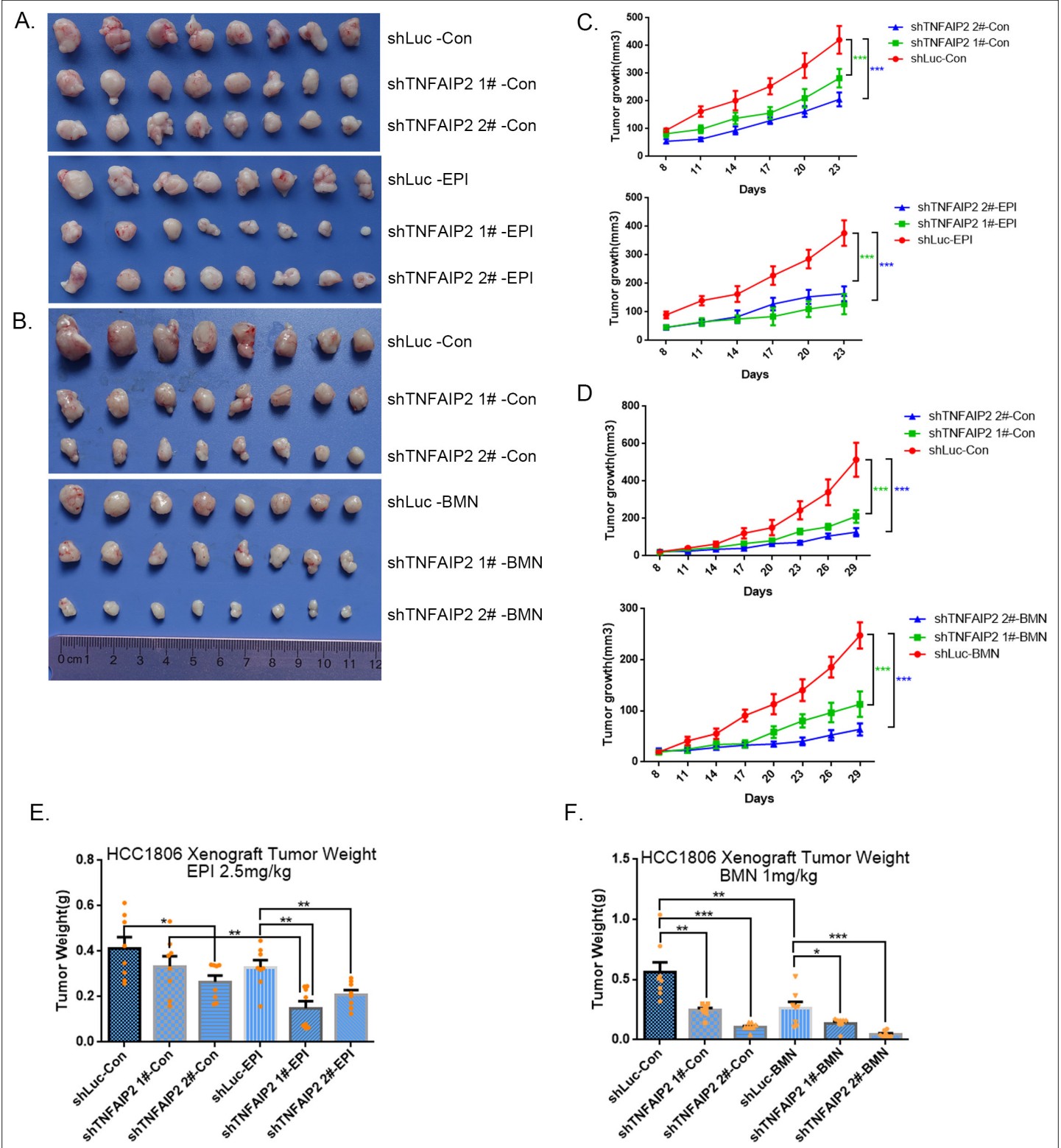

**Figure 2.** TNFAIP2 confers triple-negative breast cancer (TNBC) drug resistance in vivo. (**A–F**) TNFAIP2 knockdown increased the sensitivity ofHCC1806 breast cancer cells to EPI and BMN in vivo. HCC1806 cells with stable TNFAIP2 knockdown were transplanted into the fat pad of 7-week-old female nude mice. When the average tumor size reached approximately 50 mm³ after inoculation, mice in each group were randomly divided into two subgroups (n = 4/group) to receive EPI (2.5 mg/kg), BMN (1 mg/kg), or vehicle control for 23 or 29 days (**A, B**). Tumor size was measured twice a week (**C, D**), and tumor masses were collected and weighed at the end of the experiments (**E, F**). *p < 0.05, **p < 0.01, ***p < 0.001, t-test.

*Figure 2 continued on next page*

*Figure 2 continued*

The online version of this article includes the following source data and figure supplement(s) for figure 2:

**Figure supplement 1.** TNFAIP2 confers triple-negative breast cancer (TNBC) drug resistance in vivo.

**Figure supplement 2.** TNFAIP2 confers triple-negative breast cancer (TNBC) drug resistance in vivo.

**Figure supplement 2—source data 1.** Uncropped western blot images for *Figure 2—figure supplement 2*.

in the presence of EPI and BMN in both cell lines. Knockdown of ITGB4 also suppressed HCC1806 xenograft growth in vivo. The knockdown effect of ITGB4 protein in animal experiments was confirmed by WB (*Figure 5—figure supplement 2N*). More importantly, ITGB4 knockdown further decreased tumor volume when mice were treated with EPI and BMN (*Figure 5J–N*). Meanwhile, BMN treatment had no effect on the body weight of mice, but EPI treatment decreased mouse body weight due to its toxicity (*Figure 5—figure supplement 1H*). We then examined the effects of ITGB4 on DNA damage repair and found that ITGB4 promotes DNA damage repair in response to EPI and BMN. ITGB4 knockdown increased γH2AX and cleaved-PARP protein expression levels when HCC1806 and HCC1937 cells were treated with EPI and BMN (*Figure 5O*). Furthermore, the function of ITGB4 was validated by using two other drugs, DDP and AZD (*Figure 5—figure supplement 1A–G*). These results suggested that ITGB4 increases TNBC drug resistance and promotes DNA damage repair.

## ITGB4 activates RAC1 through TNFAIP2 and IQGAP1

It is well known that ITGB4 can activate RAC1 (*Hamill et al., 2009*) and that TNFAIP2 interacts with RAC1 and activates it (*Jia et al., 2016*). To test whether ITGB4 activates RAC1 through TNFAIP2, we measured the levels of GTP-bound RAC1 in ITGB4-overexpressing and ITGB4-knockdown cells. Overexpression of ITGB4 significantly increased the levels of GTP-bound RAC1 in both HCC1806 and HCC1937 cells (*Figure 6A*). In agreement with this observation, knockdown of ITGB4 significantly decreased the levels of GTP-bound RAC1 in both cell lines (*Figure 6B*). Next, we knocked down TNFAIP2 in ITGB4-overexpressing HCC1806 and HCC1937 cells and found that ITGB4-increased RAC1 activity was blocked by TNFAIP2 knockdown (*Figure 6C, D*). Collectively, these results demonstrate that ITGB4 activates RAC1 through TNFAIP2.

It has been reported that RAC1 activity is promoted by IQGAP1 (*Schmidt, 2012*) and that TNFAIP2 activates RAC1 through IQGAP1 (*Figure 4P*). We wondered whether ITGB4 activates RAC1 through IQGAP1; therefore, we knocked down IQGAP1 in HCC1806 and HCC1937 cells with stable overexpression of ITGB4 and found that the ITGB4-induced increase in RAC1 activity was abolished by IQGAP1 knockdown (*Figure 6E, F*). These results suggest that ITGB4 activates RAC1 through TNFAIP2 and IQGAP1.

## ITGB4 promotes TNBC drug resistance via TNFAIP2/IQGAP1/RAC1

Since ITGB4, TNFAIP2, and IQGAP1 promote drug resistance by promoting DNA damage repair in TNBC, we wondered whether ITGB4 promoted drug resistance through the TNFAIP2/IQGAP1/RAC1 axis. We knocked down TNFAIP2, IQGAP1, and RAC1 in ITGB4-overexpressing cells and found that blocking the TNFAIP2/IQGAP1/RAC1 axis increased the sensitivity of ITGB4-overexpressing HCC1806 (*Figure 7A–I*) and HCC1937 cells to EPI and BMN (*Figure 7—figure supplement 2O–W*). We also found that γH2AX and cleaved-PARP levels were upregulated in TNFAIP2/IQGAP1/RAC1 knockdown HCC1806 and HCC1937 cells stably expressing ITGB4 in the presence of EPI and BMN (*Figure 7J–L*, *Figure 7—figure supplement 2X–Z*). DDP and AZD treatment generated similar results (*Figure 7—figure supplement 1A–N*). Together, these results suggest that ITGB4 promotes DNA damage repair and drug resistance via the TNFAIP2/IQGAP1/RAC1 axis.

## TNFAIP2 expression levels positively correlated with ITGB4 in TNBC tissues

To test whether ITGB4 and TNFAIP2 are co-expressed in TNBC, we collected 135 TNBC specimens for immunohistochemistry (IHC) (the IQGAP1 antibody did not work for IHC). Specimens were obtained from the Department of Pathology, Henan Provincial People's Hospital, Zhengzhou University, China. We performed IHC analyses on two breast cancer tissue chips containing a total of 135 patients

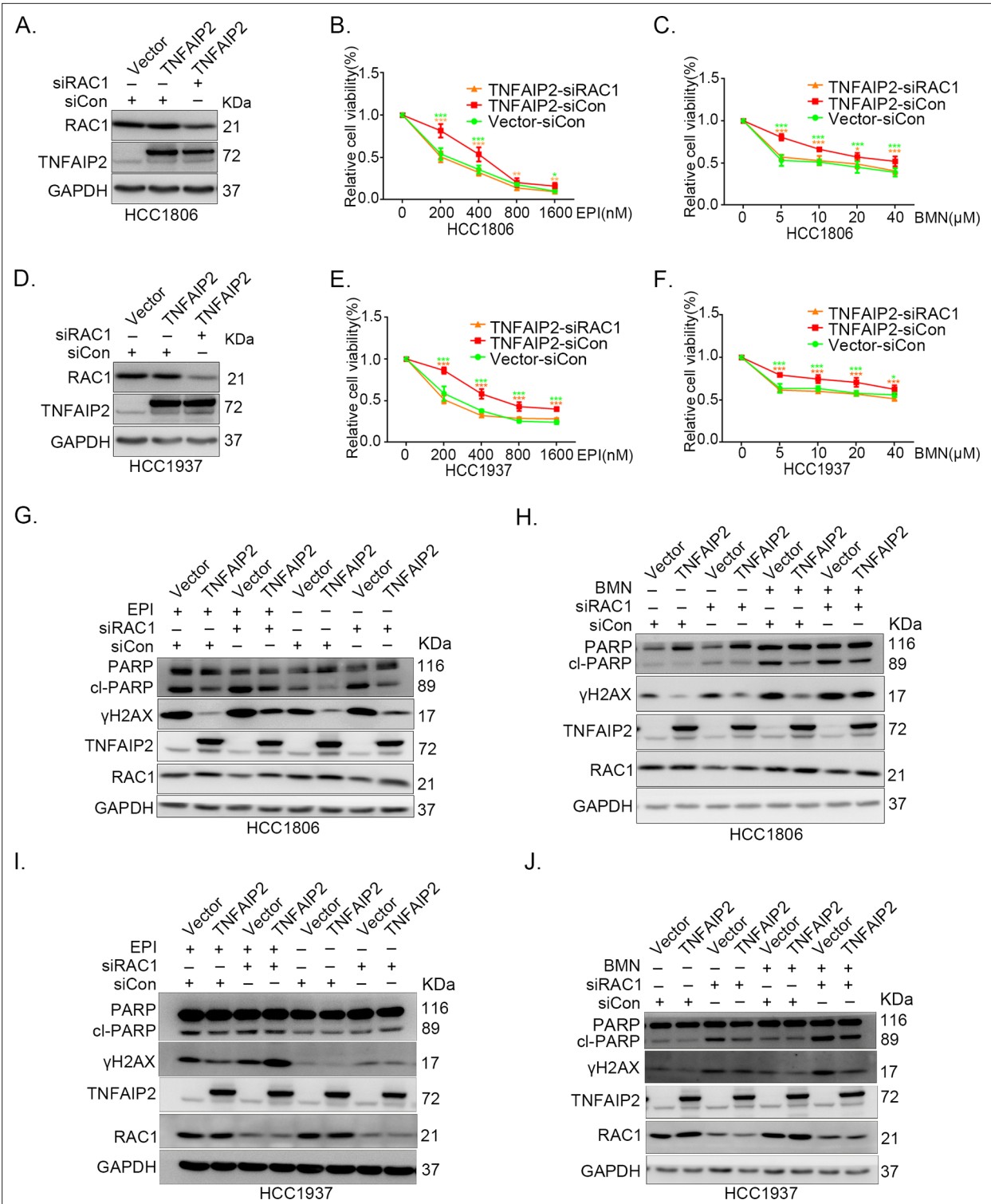

**Figure 3.** TNFAIP2 promotes triple-negative breast cancer (TNBC) drug resistance and DNA damage repair via RAC1. RAC1 knockdown abolished TNFAIP2-induced TNBC resistance to EPI and BMN. HCC1806 (**A–C**) and HCC1937 (**D–F**) cells with stable TNFAIP2 overexpression were transfected with RAC1 or control siRNA, followed by treatment with EPI (0–1600 nM) and BMN (0–40 µM) for 48 or 72 hr, respectively. Cell viability was measured by the SRB assay. Statistical analysis was performed using one-way analysis of variance (ANOVA), $n$ = 3–9, *p < 0.05, **p < 0.01, ***p < 0.001. Protein expression levels were analyzed by western blotting (WB). RAC1 depletion abolished TNFAIP2-induced DNA damage decrease in response to EPI and BMN. HCC1806 (**G–H**) and HCC1937 (**I–J**) cells with stable TNFAIP2 overexpression were transfected with RAC1 or control siRNA, followed by treatment with EPI (400 or 800 nM) and BMN (10 µM) for 24 hr, respectively. Protein expression levels were analyzed by WB.

*Figure 3 continued on next page*

*Figure 3 continued*

The online version of this article includes the following source data and figure supplement(s) for figure 3:

**Source data 1.** Uncropped western blot images for *Figure 3*.

**Figure supplement 1.** TNFAIP2 promotes triple-negative breast cancer (TNBC) drug resistance and DNA damage repair via RAC1.

**Figure supplement 1—source data 1.** Uncropped western blot images for *Figure 3—figure supplement 1*.

with TNBC (*Figure 8A–D*). TNFAIP2 and ITGB4 protein expression levels were significantly positively correlated (*Figure 8E*).

## Discussion

Chemotherapies, including EPI and DDP, are the main choice for TNBC patients. Unfortunately, TNBC frequently develops resistance to chemotherapy (*Kim et al., 2018*). Currently, PARP inhibitors are effective for TNBC with BRCA1/2 mutation or homologous recombination deficiency (HRD) (*Noordermeer and van Attikum, 2019*; *Geenen et al., 2018*; *Lee and Djamgoz, 2018*). PARP inhibitors can cause DNA damage repair defects and have synergistic lethal effects with HRD. Meanwhile, chemotherapy and PARP inhibitor resistance is also a major problem in the clinic.

In this study, we first found that TNFAIP2 promotes TNBC drug resistance and DNA damage repair through RAC1. Next, we found that TNFAIP2 interacts with IQGAP1and ITGB4. We verified that ITGB4 promotes TNBC drug resistance and DNA damage repair through the TNFAIP2/IQGAP1/RAC1 axis. Interestingly, we discovered for the first time that ITGB4 and TNFAIP2 promote RAC1 activity through IQGAP1. Our study reveals that ITGB4 promotes TNBC resistance through TNFAIP2-, IQGAP1-, and RAC1-mediated DNA damage repair (*Figure 7*). This study provides new targets for reversing TNBC resistance.

ITGB4 is well known to promote breast cancer stemness and can be activated by laminin-5 (*Campbell et al., 2018*). In addition, ITGB4 is generally in partner with ITGA6, which is another marker of breast CSCs (*Ali et al., 2011*) and drug resistance (*Campbell et al., 2018*). Therefore, whether ITGA6 has similar functions needs further study. It was reported that ITGB4 activates RAC1 (*Friedland et al., 2007*), but the mechanism is unclear. For the first time, we revealed that ITGB4 activates RAC1 through TNFAIP2 and IQGAP1. More importantly, ITGB4 promotes drug resistance through the TNFAIP2/IQGAP1/RAC1 axis.

TNFAIP2 plays important roles in different cellular and physiological processes, including cell proliferation, adhesion, migration, membrane TNT formation, angiogenesis, inflammation, and tumorigenesis (*Jia et al., 2018*). We previously found that TNFAIP2 was regulated by KLF5 and interacted with the small GTPases RAC1 and CDC42, thereby regulating the actin cytoskeleton and cell morphology in breast cancer cells (*Jia et al., 2016*). However, the detailed mechanism is not clearly understood. In this study, we found that IQGAP1 mediates this process. IQGAP1 is a crucial regulator of cancer development by scaffolding and facilitating different oncogenic pathways, especially RAC1/CDC42, thus affecting proliferation, adhesion, migration, invasion, and metastasis (*Wei and Lambert, 2021*). In addition, IQGAP1 is increased during the differentiation of ovarian CSCs and promotes aggressive behaviors (*Huang et al., 2015*). In our study, we found that TNFAIP2 interacts with IQGAP1 and thus activates RAC1 to induce chemotherapy and PARP inhibitor drug resistance.

Furthermore, TNFAIP2 was reported to induce epithelial-to-mesenchymal transition and confer platinum resistance in urothelial cancer cells (*Niwa et al., 2019*), and in embryonic stem cell (ESC) differentiation, TNFAIP2 was found to be important in controlling lipid metabolism, which supports the ESC differentiation process and planarian organ maintenance (*Deb et al., 2021*). Another study found that TNFAIP2 overexpression enhanced TNT-mediated autophagosome and lysosome exchange, preventing advanced glycation end product (AGE)-induced autophagy and lysosome dysfunction and apoptosis (*Barutta et al., 2023*). In cancer treatment, TNFAIP2 was chosen as one of the six signature genes predicting chemotherapeutic and immunotherapeutic efficacies, with high-senescore patients benefiting from immunotherapy and low-senescore patients responsive to chemotherapy (*Zhou et al., 2022*).

These reports provide a possible explanation for previous studies showing that ITGB4 is important in EMT and cancer stemness. According to our results that there is an interaction between ITGB4

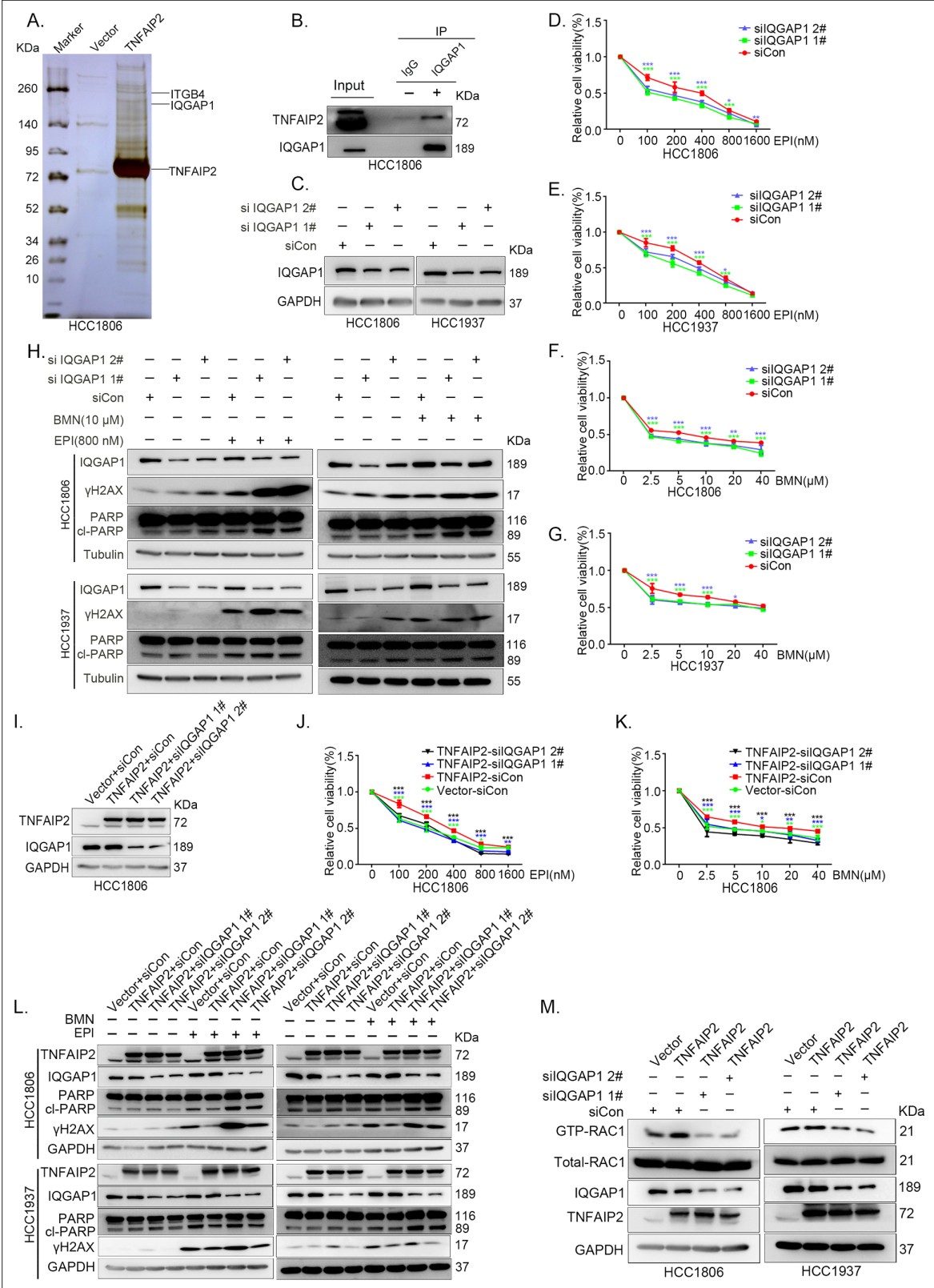

**Figure 4.** IQGAP1 mediates RAC1 activation by TNFAIP2 and promotes triple-negative breast cancer (TNBC) drug resistance. (**A**) The IP-MS result of TNFAIP2 in HCC1806 cells. (**B**) Endogenous TNFAIP2 interacts with IQGAP1 in HCC1806 cells. Endogenous TNFAIP2 protein was immunoprecipitated using an anti-IQGAP1 antibody. Immunoglobulin (Ig)G served as the negative control. Endogenous TNFAIP2 was detected by western blotting (WB). (**C–G**) IQGAP1 knockdown in HCC1806 and HCC1937 cells significantly decreased cell viability in the presence of EPI (0–1600 nM) and BMN (0–40 μM), as

*Figure 4 continued on next page*

*Figure 4 continued*

measured by the SRB assay. Statistical analysis was performed using one-way analysis of variance (ANOVA), n = 3–6, *p < 0.05, **p < 0.01, ***p < 0.001. IQGAP1 protein expression was detected by WB. (**H**) IQGAP1 knockdown in HCC1806 and HCC1937 cells increased DNA damage of EPI and BMN. HCC1806 and HCC1937 cells with IQGAP1 knockdown were treated with 800 nM EPI for 24 hr and 10 μM BMN for 24 hr, respectively. ITGB4, γH2AX, and PARP protein expression was detected by WB. (**I–K**) IQGAP1 knockdown abolished TNFAIP2-confered resistance to EPI and BMN. HCC1806 cells with stable TNFAIP2 overexpression were transfected with IQGAP1 or control siRNA, followed by treatment with EPI (0–1600 nM) and BMN (0–40 μM) for 48 or 72 hr, respectively. Cell viability was measured by the SRB assay. Statistical analysis was performed using one-way ANOVA, n = 3, *p < 0.05, **p < 0.01, ***p < 0.001. IQGAP1 protein expression was detected by WB. (**L**) IQGAP1 knockdown abolished TNFAIP2-confered resistance to EPI and BMN. HCC1806 and HCC1937 cells with stable TNFAIP2 overexpression were transfected with IQGAP1 or control siRNA, followed by treatment with EPI (800 nM) and BMN (10 μM) for 24 hr, respectively. Protein expression levels were analyzed by WB. (**M**) IQGAP1 knockdown abolished TNFAIP2-confered RAC1 activation. HCC1806 and HCC1937 cells with stable TNFAIP2 overexpression were transfected with IQGAP1 or control siRNA. GTP-RAC1 levels were assessed using PAK-PBD beads.

The online version of this article includes the following source data and figure supplement(s) for figure 4:

**Source data 1.** Uncropped western blot images for *Figure 4*.

**Figure supplement 1.** IQGAP1 mediates RAC1 activation by TNFAIP2 and promotes triple-negative breast cancer (TNBC) drug resistance.

**Figure supplement 1—source data 1.** Uncropped western blot images for *Figure 4—figure supplement 1*.

and TNFAIP2, ITGB4 might regulate EMT and stemness through TNFAIP2. TNFAIP2 is one of the important factors induced by tumor necrosis factor alpha (TNFα). Interestingly, TNFα release could be induced by therapeutic drugs from multiple tumor cell lines. The acquisition of docetaxel resistance was accompanied by increased constitutive production of TNFα (*Guo and Yuan, 2020*). In addition, TNFα is a key tumor-promoting effector molecule secreted by tumor-associated macrophages. In vitro neutralizing TNFα was observed to inhibit tumor progression and improve the curative effect of bevacizumab (*Liu et al., 2020*). Therefore, the mechanism by which TNFα promotes chemotherapeutic resistance in breast cancer should be further investigated.

For future studies, it will be important to develop *Tnfaip2* knockout mice to investigate the exact role of TNFAIP2 physiologically. According to recent studies and our findings, agents targeting the interaction among ITGB4/TNFAIP2/IQGAP1 would be a promising trend for developing drugs to overcome the resistance phenomenon.

In summary, ITGB4 and TNFAIP2 play important roles in breast cancer chemoresistance. TNFAIP2 activates RAC1 to promote chemoresistance through IQGAP1. In addition, ITGB4 activates RAC1 through TNFAIP2 and IQGAP1 and confer DNA damage-related drug resistance in TNBC (*Figure 8F*). These results indicate that the ITGB4/TNFAIP2/IQGAP1/RAC1 axis provides potential therapeutic targets to overcome DNA damage-related drug resistance in TNBC.

## Materials and methods
### Cell lines and reagents

All cell lines used in this study, including HCC1806, HCC1937, and HEK293T cells, were purchased from ATCC (American Type Culture Collection, Manassas, VA, USA) and validated by STR (short tandem repeat) analysis and these cell lines tested negative for mycoplasma contamination. HCC1806 and HCC1937 cells were cultured in RPMI 1640 medium supplemented with 5% fetal bovine serum (FBS). HEK293T cells were cultured in DMEM (Thermo Fisher, Grand Island, USA) with 5% FBS at 37°C with 5% $CO_2$. Epirubicin (EPI) (Cat#HY-13624A), cisplatin (DDP) (Cat#HY-17394), talazoparib (BMN) (Cat#HY-16106), and olaparib (AZD) (Cat#HY-10162) were purchased from MCE (New Jersey, USA).

### Plasmid construction and stable TNFAIP2 and ITGB4 overexpression

We constructed the full-length *TNFAIP2/ITGB4* gene and then subcloned them into the pCDH lentiviral vector. The packaging plasmids (including pMDLg/pRRE, pRSV-Rev, and pCMV-VSV-G) and pCDH-TNFAIP2/ITGB4 expression plasmid were cotransfected into HEK293T cells (2 × 10⁶ in 10 cm plate) to produce lentivirus. Following transfection for 48 hr, the lentivirus was collected and used to infect HCC1806 and HCC1937 cells. Forty-eight hours later, puromycin (2 μg/ml) was used to screen the cell populations.

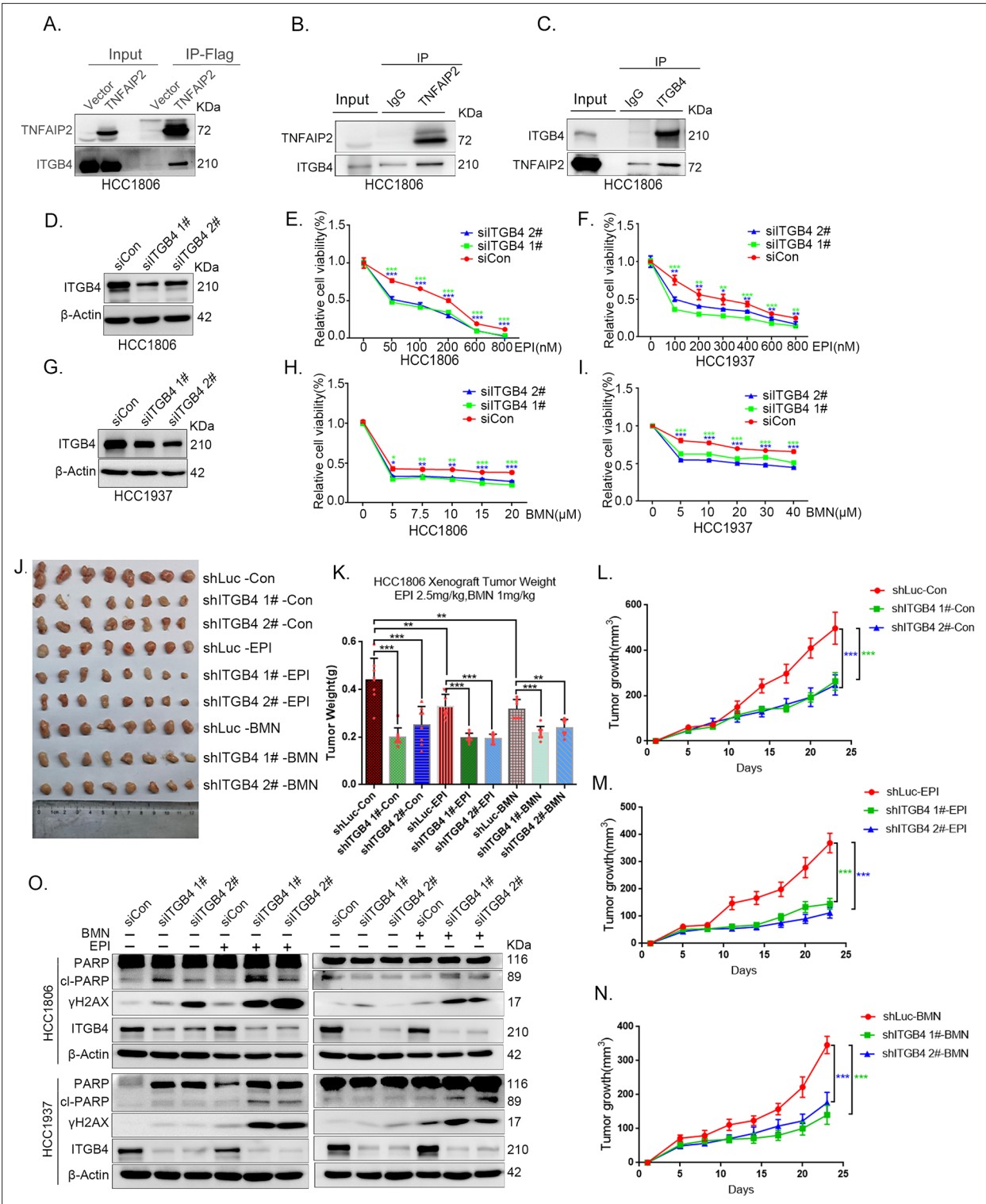

**Figure 5.** ITGB4 interacts with TNFAIP2 and promotes triple-negative breast cancer (TNBC) drug resistance and DNA damage repair. (**A**) TNFAIP2 interacts with ITGB4.HCC1806 cells with stable TNFAIP2 overexpression were collected from flag-tagged TNFAIP2 cell lysates for immunoprecipitation assays using Flag-M2 beads, and ITGB4 was detected by western blotting (WB). (**B**) Endogenous TNFAIP2 interacts with ITGB4 in HCC1806 cells. Endogenous TNFAIP2 protein was immunoprecipitated using an anti-TNFAIP2 antibody. IgG served as the negative control. Endogenous ITGB4 was detected by WB. (**C**) Endogenous ITGB4 interacts with TNFAIP2 in HCC1806 cells. Endogenous ITGB4 protein was immunoprecipitated using an anti-ITGB4 antibody. IgG served as the negative control. Endogenous TNFAIP2 was detected by WB. (**D–I**) ITGB4 knockdown in HCC1806 and HCC1937 cells significantly decreased cell viability in the presence of EPI (0–800 nM) and BMN (0–40 μM), as measured by the SRB assay. Statistical analysis was

*Figure 5 continued on next page*

performed using one-way analysis of variance (ANOVA), n = 3, *p < 0.05, **p < 0.01, ***p < 0.001. ITGB4 protein expression was detected by WB. (**J–N**) ITGB4 depletion promotes HCC1806 breast cancer cell sensitivity to EPI and BMN treatment in vivo. HCC1806 cells with stable ITGB4 knockdown were transplanted into the fat pad of 7-week-old female nude mice. When the average tumor size reached approximately 50 mm³ after inoculation, the mice in each group were randomly divided into two subgroups (n = 4/group) to receive EPI (2.5 mg/kg), BMN (1 mg/kg), or vehicle control for 22 days (**J**). Tumor masses were collected and weighed at the end of the experiments (**K**), and tumor size was measured twice a week (**L–N**). *p < 0.05, **p < 0.01, ***p < 0.001, t-test. (**O**) ITGB4 knockdown increased DNA damage of EPI and BMN. HCC1806 and HCC1937 cells with ITGB4 knockdown were treated with 400 nM EPI for 24 hr and 5 µM BMN for 24 hr, respectively. ITGB4, γH2AX, and PARP protein expression was detected by WB.

The online version of this article includes the following source data and figure supplement(s) for figure 5:

**Source data 1.** Uncropped western blot images for *Figure 5*.

**Figure supplement 1.** ITGB4 interacts with TNFAIP2 and promotes triple-negative breast cancer (TNBC) drug resistance and DNA damage repair.

**Figure supplement 1—source data 1.** Uncropped western blot images for *Figure 5—figure supplement 1*.

**Figure supplement 2.** ITGB4 interacts with TNFAIP2 and promotes triple-negative breast cancer (TNBC) drug resistance and DNA damage repair.

**Figure supplement 2—source data 1.** Uncropped western blot images for *Figure 5—figure supplement 2*.

## Stable knockdown of TNFAIP2 and ITGB4

The pSIH1-H1-puro shRNA vector was used to express TNFAIP2, ITGB4, and luciferase (LUC) shRNAs. *TNFAIP2*shRNA#1, 5'-GACUUGGGCUCACAGAUAA-3'; *TNFAIP2*shRNA#2, 5'-GAUUGAGGUGGC-CACUUAU-3'; *ITGB4*shRNA#1, 5'-ACGACAGCTTCCTTATGTA-3'; *ITGB4*shRNA#2, 5'-CAGCGACT ACACTATTGGA-3'; *Luciferase*shRNA, 5'-CUUACGCUGAGUACUUCGA-3'; HCC1806 and HCC1937 cells were infected with lentivirus. Stable populations were selected using 1–2 mg/ml puromycin. The knockdown effect was evaluated by WB.

## RNA interference

The siRNA target sequences used in this study are as follows:*TNFAIP2*siRNA#1, 5'-GACUUGGGCUCA CAGAUAA-3'; *TNFAIP2*siRNA#2, 5'-GAUUGAGGUGGCCACUUAU-3'; *ITGB4*siRNA#1, 5'-ACGA CAGCTTCCTTATGTA-3'; *ITGB4*siRNA#2, 5'-CAGCGACTACACTATTGGA-3'; *RAC1*siRNA, 5'-CGGC ACCACUGUCCCAACA-3'; *IQGAP1*siRNA#1, 5'-GCAGGTGGATTACTATAAA-3'; *IQGAP1*siRNA#2, 5'-CUAGUGAAACUGCAACAGA-3'. All siRNAs were synthesized by RiboBio (RiboBio, China) and transfected at a final concentration of 50 nM.

## Antibodies and WB

The WB procedure has been described in our previous study (*Chen et al., 2005*). Anti-TNFAIP2 (sc-28318), anti-ITGB4 (sc-9090), and anti-GAPDH (sc-25778) antibodies were purchased from Santa Cruz Biotechnology (Santa Cruz, CA, USA). The anti-PARP (#9542) antibody was purchased from CST. Anti-RAC1 (05-389) and anti-γH2AX (3475627) antibodies were purchased from Millipore (Billerica, MA, USA). Anti-β-actin (A5441) and anti-Tubulin (T5168) antibodies were purchased from Sigma-Aldrich (St Louis, MO, USA). The anti-IQGAP1 (ab86064) antibody was purchased from Abcam.

## Immunoprecipitation and silver staining

Immunoprecipitation and silver staining lysates from HCC1806 cells stably expressing Flag-TNFAIP2 were prepared by incubating the cells in lysis buffer containing a protease inhibitor cocktail (MCE). Cell lysates were obtained from approximately $2.5 \times 10^8$ cells, and after binding with anti-Flag M2 affinity gel (Sigma) for 2 hr as recommended by the manufacturer, the affinity gel was washed with cold lysis plus 0.2% NP-40. FLAG peptide (Sigma) was applied to elute the Flag-labeled protein complex as described by the vendor. The elutes were collected and visualized on NuPAGE 4–12% Bis-Trisgels (Invitrogen, CA, USA) followed by silver staining with a silver staining kit (Pierce, IL, USA). The distinct protein bands were retrieved and analyzed by Liquid Chromatograph-Mass Spectrometer (LC–mass).

## Immunoprecipitation and GST pull-down

For exogenous interaction between ITGB4 and Flag-TNFAIP2, cell lysates were directly incubated with anti-Flag M2 affinity gel (A2220; Sigm) overnight at 4°C. For endogenous protein interaction, cell lysates were first incubated with anti-TNFAIP2/ITGB4/IQGAP1 antibodies or mouse IgG/rabbit IgG (sc-2028; Santa Cruz Biotechnology, CA, USA) and then incubated with Protein A/G plusagarose

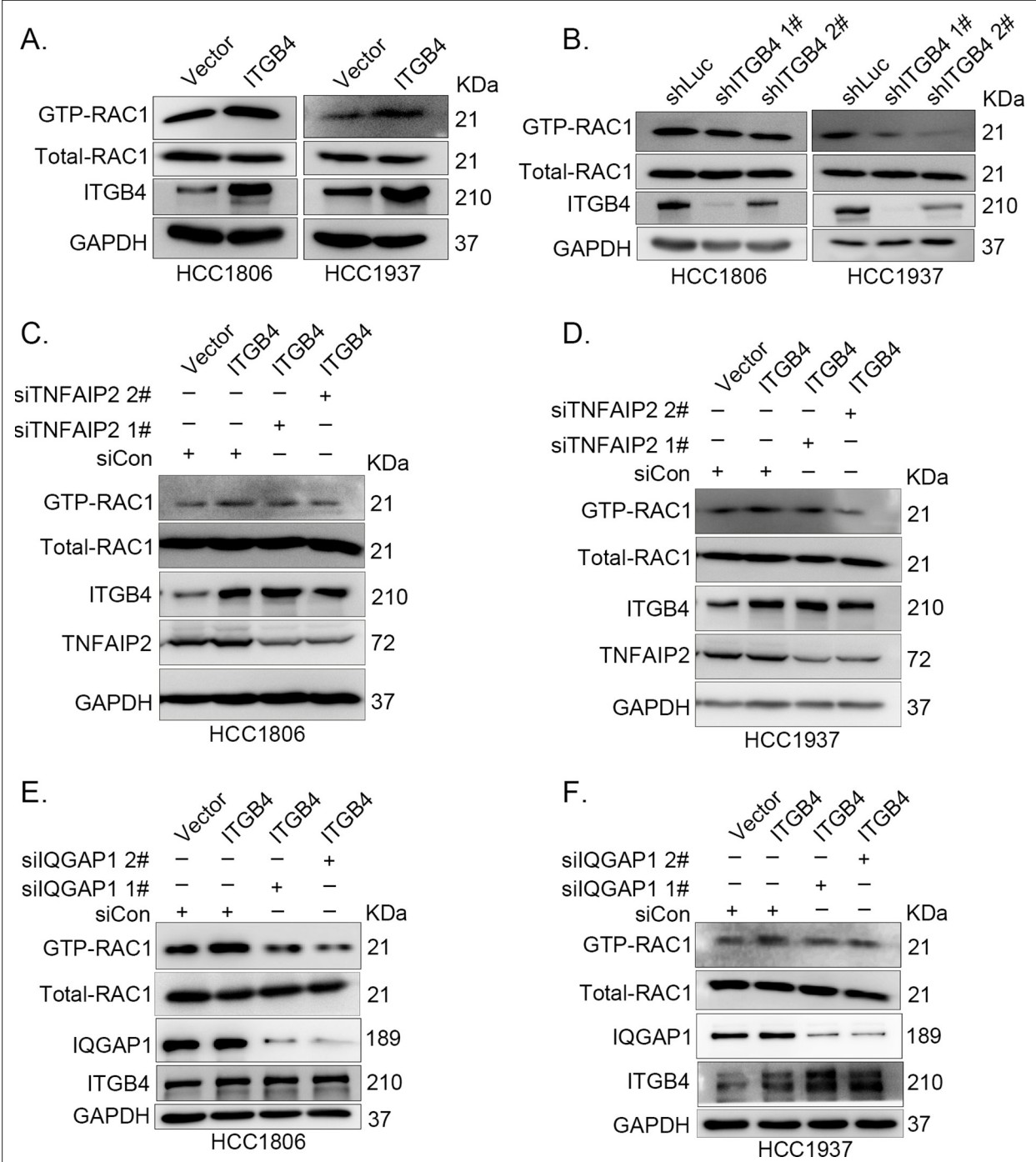

**Figure 6.** ITGB4 activates RAC1 through TNFAIP2 and IQGAP1. (**A**) Overexpression of ITGB4-increased GTP-RAC1 levels in HCC1806 and HCC1937 cells. GTP-RAC1 were assessed using PAK-PBD beads. (**B**) Knockdown of ITGB4 by shRNA decreased GTP-RAC1 levels in HCC1806 and HCC1937 cells. ITGB4 activates RAC1 through TNFAIP2. HCC1806 (**C**) and HCC1937 (**D**) cells with stable ITGB4 overexpression were transfected with TNFAIP2 or control siRNA. ITGB4 activates RAC1 through IQGAP1. HCC1806 (**E**) and HCC1937 (**F**) cells with stable ITGB4 overexpression were transfected with IQGAP1 or control siRNA.

The online version of this article includes the following source data for figure 6:

**Source data 1.** Uncropped western blot images for *Figure 6*.

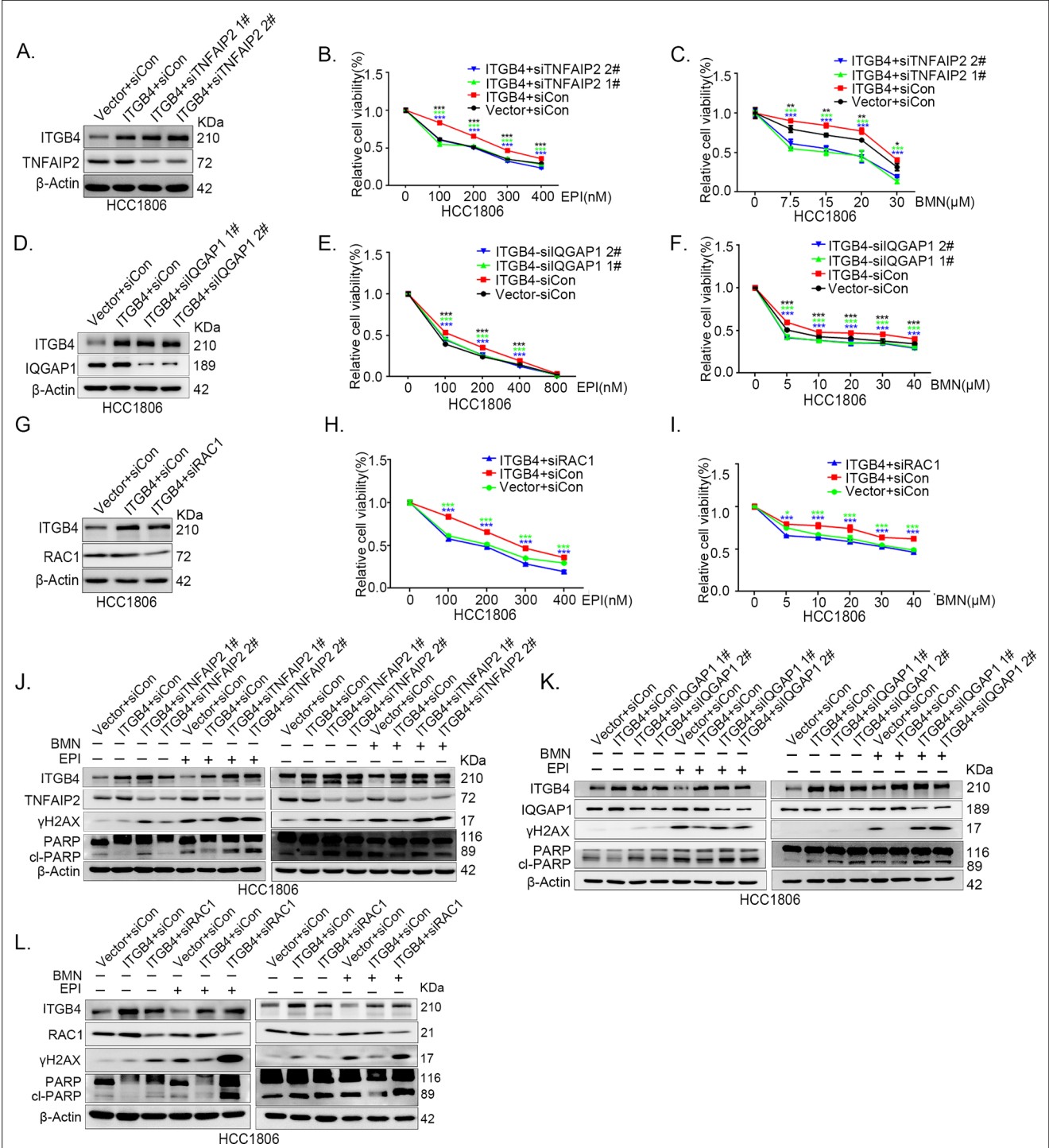

**Figure 7.** ITGB4 promotes triple-negative breast cancer (TNBC) drug resistance via TNFAIP2/IQGAP1/RAC1. (**A–C**) ITGB4 promotes TNBC drug resistance through TNFAIP2. TNFAIP2 knockdown abolished ITGB4-induced resistance to EPI and BMN. HCC1806 cells with stable ITGB4 overexpression were transfected with TNFAIP2 or control siRNA, followed by treatment with EPI (0–400 nM) and BMN (0–30 μM) for 48 or 72 hr, respectively. Cell viability was measured by the SRB assay. Statistical analysis was performed using one-way analysis of variance (ANOVA), $n = 3$, *$p <$ 0.05, **$p <$ 0.01, ***$p <$ 0.001. Protein expression levels were analyzed by western blotting (WB). (**D–F**) ITGB4 promotes TNBC drug resistance through IQGAP1. HCC1806 cells with stable ITGB4 overexpression were transfected with IQGAP1 or control siRNA, followed by treatment with EPI (0–800 nM) and BMN (0–40 μM) for 48 or 72 hr, respectively. Cell viability was measured by the SRB assay. Statistical analysis was performed using one-way ANOVA, $n = 3$, *$p <$ 0.05, **$p <$ 0.01, ***$p <$ 0.001. Protein expression levels were analyzed by WB. (**G–I**) ITGB4 promotes TNBC drug resistance through RAC1. HCC1806 cells with stable ITGB4 overexpression were transfected with RAC1 or control siRNA, followed by treatment with EPI (0–400 nM) and BMN

*Figure 7 continued on next page*

*Figure 7 continued*

(0–40 µM) for 48 or 72 hr, respectively. Cell viability was measured by the SRB assay. Statistical analysis was performed using one-way ANOVA, $n = 3$, *$p < 0.05$, **$p < 0.01$, ***$p < 0.001$. Protein expression levels were analyzed by WB. (**J**) ITGB4 promotes DNA damage repair through TNFAIP2. HCC1806 cells with stable ITGB4 overexpression were transfected with TNFAIP2 or control siRNA, followed by treatment with EPI (400 nM) and BMN (5 µM) for 24 hr. Protein expression levels were analyzed by WB. (**K**) ITGB4 promotes DNA damage repair through IQGAP1. HCC1806 cells with stable ITGB4 overexpression were transfected with IQGAP1 or control siRNA, followed by treatment with EPI (400 nM) and BMN (5 µM) for 24 hr. Protein expression levels were analyzed by WB. (**L**) ITGB4 promotes DNA damage repair through RAC1. HCC1806 cells with stable ITGB4 overexpression were transfected with RAC1 or control siRNA, followed by treatment with EPI (400 nM) and BMN (5 µM) for 24 hr. Protein expression levels were analyzed by WB.

The online version of this article includes the following source data and figure supplement(s) for figure 7:

**Source data 1.** Uncropped western blot images for *Figure 7*.

**Figure supplement 1.** ITGB4 promotes triple-negative breast cancer (TNBC) drug resistance via TNFAIP2/IQGAP1/RAC1.

**Figure supplement 1—source data 1.** Uncropped western blot images for *Figure 7—figure supplement 1*.

**Figure supplement 2.** ITGB4 promotes triple-negative breast cancer (TNBC) drug resistance via TNFAIP2/IQGAP1/RAC1.

**Figure supplement 2—source data 1.** Uncropped western blot images for *Figure 7—figure supplement 2*.

beads (sc-2003; Santa Cruz Biotechnology). For the GST pull-down assay, cell lysates were directly incubated with GlutathioneSepharose 4B (52-2303-00; GE Healthcare) overnight at 4°C. The precipitates were washed four times with 1 ml of lysis buffer, boiled for 10 min with 1× sodium dodecyl sulfate (SDS) sample buffer, and subjected to WB analysis.

## Cell viability assays

Cell viability was measured by SRB assays as described in our previous study (*Chen et al., 2007*). Cell viability was measured by SRB assays. Briefly, cells were seeded in 96-well plates. Then, the cells were cultured for the indicated time and fixed with 10% trichloroacetic acid at room temperature for 30 min, followed by incubation with 0.4% SRB (wt/vol) solution in 1% acetic acid for 20 min at room temperature. Finally, SRB was dissolved in 10 mM unbuffered Tris base, and the absorbance was measured at a wavelength of 530 nm on a plate reader (Bio Tek, Vermont, USA).

## RAC1 activation assays

RAC1 activation was examined using the Cdc42 Activation Assay Biochem Kit (BK034, Cytoskeleton, Denver, USA) following the manufacturer's instructions. Cells were harvested with cell lysis buffer, and1 mg of protein lysate in a 1 ml total volume at 4°C was immediately precipitated with 10 µg of PAK-PBD beads for 60 min with rotation. After washing three times with wash buffer, agarose beads were resuspended in 30 µl of 2× SDS sample buffer and boiled for 5 min. RAC1-GTP was examined by WB with an anti-RAC1 antibody.

## Xenograft experiments

We purchased 6- to 7-week-old female BALB/cnude mice from SLACCAS (Changsha, China). HCC1806-shLuc, HCC1806-shTNFAIP2, or HCC1806-shITGB4 cells ($1 \times 10^6$ in Matrigel (BD Biosciences, NY, USA)) were implanted into the mammary fat pads of the mice. When the tumor volume reached approximately 50 mm³, the nude mice were randomly assigned to the control and treatment groups ($n = 4$/group). EPI, BMN, and DDP were dissolved in ddH₂O. The control group was given vehicle alone, and the treatment group received EPI (2.5 mg/kg), BMN (1 mg/kg), and DDP (2.5 mg/kg) alone via intraperitoneal injection every 3 days for 18 or 27 days. The tumor volume was calculated as follows: tumor volume was calculated by the formula: $(\pi \times \text{length} \times \text{width}^2)/6$.

## Immunohistochemical staining

Paraffin-embedded clinical TNBC specimens were obtained from the Department of Pathology, Henan Provincial People's Hospital, Zhengzhou University, Henan, China. Two tissue microarrays containing 135 TNBC breast cancer tissues were constructed. For the IHC assay, the slides were deparaffinized, rehydrated, and pressure cooker heated for 2.5 min in EDTA for antigen retrieval. Endogenous peroxidase activity was inactivated by adding an endogenous peroxidase blocker (OriGene, China) for 15 min at room temperature. Slides were incubated overnight at 4°C with anti-TNFAIP2 (1:200) or anti-ITGB4 (1:500). After 12 hr, the slides were washed three times with PBS and incubated with secondary antibodies

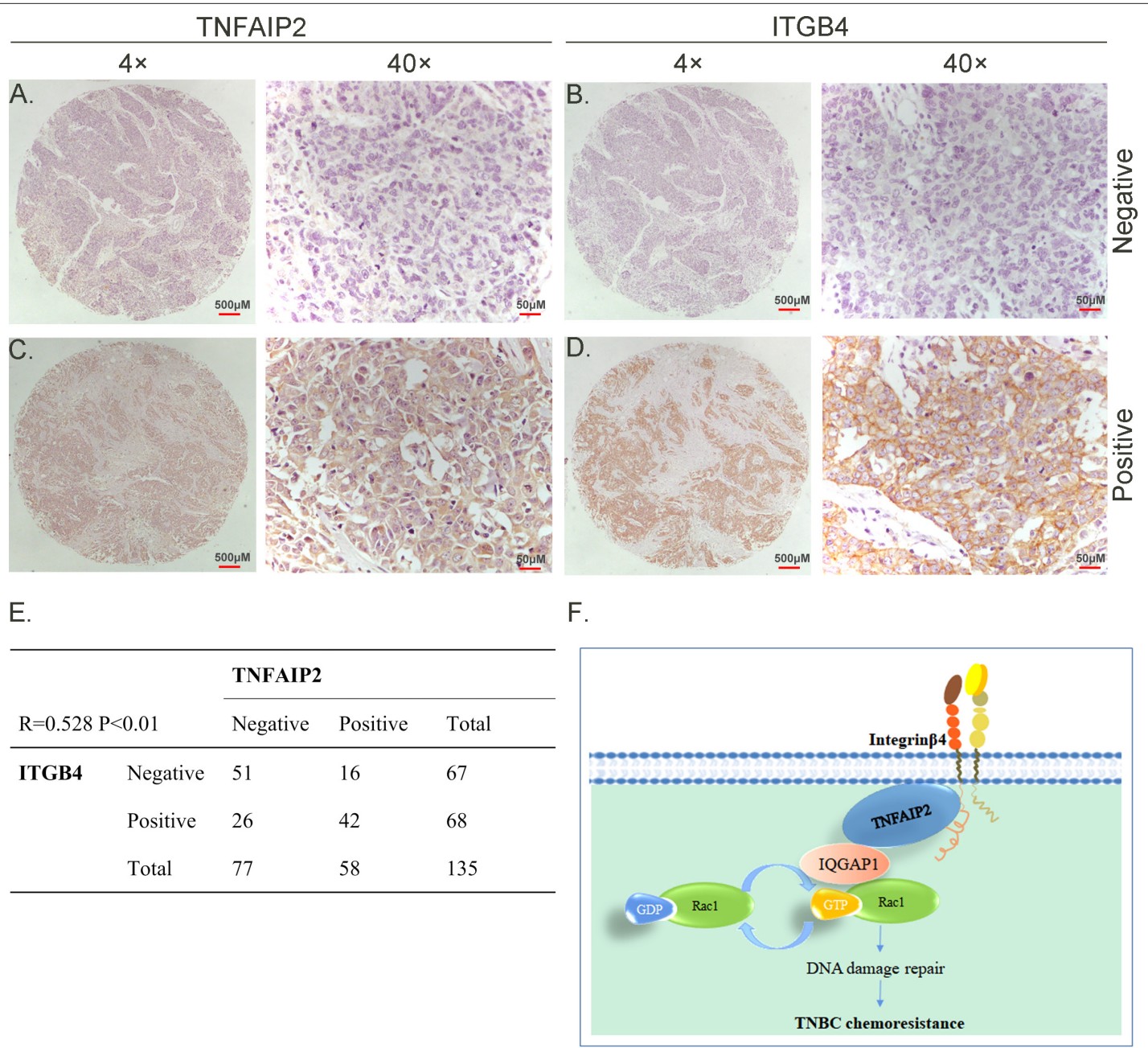

**Figure 8.** TNFAIP2 expression levels positively correlated with ITGB4 in triple-negative breast cancer (TNBC) tissues. Representative immunohistochemistry (IHC) images of TNFAIP2 and ITGB4 protein expression in breast cancer tissues are shown. The final expression assessment was performed by combining the two scores (0–2 = low, 6–7 = high). **A** and **B** indicate low scores, **C** and **D** indicate high scores, and **E** indicates that the TNFAIP2 and ITGB4 protein expression levels are positively correlated in human TNBC specimens. Figure **F** is the work model of this study.

hypersensitive enzyme-labeled goat anti-mouse/rabbit IgG polymer (OriGene, China) at room temperature for 20 min, DAB concentrate chromogenic solution (1:200dilution of concentrated DAB chromogenic solution), counterstained with 0.5% hematoxylin, dehydrated with graded concentrations of ethanol for 3 min each (70–80–90–100%), and finally stained with dimethyl benzene immunostained slides were evaluated by light microscopy. The IHC signal was scored using the 'Allred Score' method.

## Statistical analysis

All graphs were created using GraphPad Prism software version 8.0. Comparisons between two independent groups were assessed by two-tailed Student's *t*-test. One-way analysis of variance with least

significant differences was used for multiple group comparisons. p-values of <0.05, 0.01, or 0.001 were considered to indicate a statistically significant result, comparisons significant at the 0.05 level are indicated by *, at the 0.01 level are indicated by **, or at the 0.001 level are indicated by ***.

## Acknowledgements

This work was supported by National Key R&D Program of China (2020YFA0112300), National Natural Science Foundation of China (81830087, U2102203, 81672624, 82102987, and 82203413), the Yunnan Fundamental Research Projects (202101AS070050), the Guangdong Foundation Committee for Basic and Applied Basic Research projects (2022A1515012420), and Yunnan (Kunming) Academician Expert Workstation (grant no. YSZJGZZ-2020025).

## Additional information

### Funding

| Funder | Grant reference number | Author |
|---|---|---|
| National Key Research and Development Program of China | 2020YFA0112300 | Ceshi Chen |
| National Natural Science Foundation of China | 81830087 | Ceshi Chen |
| National Natural Science Foundation of China | 81672624 | Jing Feng |
| National Natural Science Foundation of China | 82102987 | Qiuxia Cui |
| National Natural Science Foundation of China | 82203413 | Huichun Liang |
| The Yunnan Fundamental Research Projects | 202101AS070050 | Ceshi Chen |
| The Guangdong Foundation Committee for Basic and Applied Basic Research projects | 2022A1515012420 | Qiuxia Cui |
| Yunnan (Kunming) Academician Expert Workstation | grant No. YSZJGZZ-2020025 | Ceshi Chen |
| National Natural Science Foundation of China | U2102203 | Ceshi Chen |

The funders had no role in study design, data collection, and interpretation, or the decision to submit the work for publication.

### Author contributions

Huan Fang, Data curation, Writing – original draft; Wenlong Ren, Chuanyu Yang, Data curation; Qiuxia Cui, Funding acquisition; Huichun Liang, Conceptualization, Writing - review and editing; Wenjing Liu, Xue Liu, Methodology; Xinye Wang, Yujie Shi, Resources; Jing Feng, Conceptualization, Funding acquisition; Ceshi Chen, Conceptualization, Supervision, Writing - review and editing

### Author ORCIDs

Qiuxia Cui ⬤ http://orcid.org/0000-0003-3635-5420
Ceshi Chen ⬤ http://orcid.org/0000-0001-6398-3516

### Ethics

Paraffin-embedded clinical TNBC specimens were obtained from the Department of Pathology, Henan Provincial People's Hospital, Zhengzhou University, Henan, China. This project has received medical ethics support (YS2021036).

Animal feeding and experiments were approved by the animal ethics committee of the affiliated Hospital of Guangdong Medical university (GDY2102096).

Reviewer #1 (Public Review): https://doi.org/10.7554/eLife.88483.3.sa1
Reviewer #2 (Public Review): https://doi.org/10.7554/eLife.88483.3.sa2
Reviewer #3 (Public Review): https://doi.org/10.7554/eLife.88483.3.sa3
Author Response https://doi.org/10.7554/eLife.88483.3.sa4

---

## Additional files

### Supplementary files
• Supplementary file 1. Protein bands.
• MDAR checklist

### Data availability
The authors confirm that the data supporting the findings of this study are available within the article and its supplementary materials.

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
