## [Editor Report · eLife assessment]

This study presents a rather **valuable** finding that IQGAP1 interacts with TNFAIP2, which activates Rac1 to promote drug resistance in TNBC. The evidence supporting the claims of the authors is quite **solid**. The work will be of interest to scientists working on breast cancer.

---

## [Referee Report · Reviewer #1 (Public Review)]

In this study, Fang H et al. describe a potential pathway, ITGB4-TNFAIP2-IQGAP1-Rac1, that may involve in the drug resistance in triple negative breast cancer (TNBC). Mechanistically, it was demonstrated that TNFAIP2 bind with IQGAP1 and ITGB4 activating Rac1 and the following drug resistance. The present study focused on breast cancer cell lines with supporting data from mouse model and patient breast cancer tissues. The study is interesting. The experiments were well controlled and carefully carried out. The conclusion is supported by strong evidence provided in the manuscript. The authors may want to discuss the link between ITGB4 and Rac 1, between IQGAP1 and Rac1, and between TNFAIP2 and Rac1 as compared with the current results obtained. This is important considering some recent publications in this area (Cancer Sci 2021, J Biol Chem 2008, Cancer Res 2023). In addition, some key points need to be addressed in order to support their conclusion in full.

---

## [Referee Report · Reviewer #2 (Public Review)]

Breast cancer is the most common malignant tumor in women. One of subtypes in breast cancer is so called triple-negative breast cancer (TNBC), which represents the most difficult subtype to treat and cure in the clinic. Chemotherapy drugs including epirubicin and cisplatin are widely used for TNBC treatment. However, drug resistance remains as a challenge in the clinic. The authors uncovered a molecular pathway involved in chemotherapy drug resistance, and molecular players in this pathway represent as potential drug targets to overcome drug resistance. The experiments are well designed and the conclusions drawn mostly were supported by the data. The findings have potential to be translated into the clinic.

---

## [Referee Report · Reviewer #3 (Public Review)]

A summary of what the authors were trying to achieve:

- TNFAIP2 promotes TNBC drug resistance and DNA damage repair.

- Mechanistically, TNFAIP2 interacts with IQGAP1 and Integrin β4 to mediate RAC1 activation and thus promotes TNBC drug resistance.

- Clinically, TNFAIP2 expression levels positively correlated with ITGB4 in TNBC tissues.

- ITGB4 and TNFAIP2 might serve as promising therapeutic targets for TNBC.

-An account of the major strengths and weaknesses of the methods and results.

The authors performed numerous rescue experiments in vitro to confirm the relationship among ITGB4, TNFAIP2, IQGAP1 and Rac1. However, clinical relevance is somehow limited. Additional experiments are needed to demonstrate the above conclusions in clinical samples.

-An appraisal of whether the authors achieved their aims, and whether the results support their conclusions.

To most extent, the authors achieved their aims, and the results demonstrate their conclusions "TNFAIP2 interacts with IQGAP1 and ITGB4. ITGB4 promotes TNBC drug resistance via the TNFAIP2/IQGAP1/RAC1 axis by promoting DNA damage repair".

-A discussion of the likely impact of the work on the field, and the utility of the methods and data to the community.

Drug resistance is always a challenge for TNBC treatment. This paper found that TNFAIP2 interacts with IQGAP1 and ITGB4 to activate Rac1, thus conferring DNA chemo-resistance to TNBC cells. In addition, positive correlation between the expression of TNFAIP2 and ITGB4 in TNBC tissues were presented. This paper suggests that the ITGB4/TNFAIP2/IQGAP1/Rac1 axis provides potential therapeutic targets to overcome chemo-resistance (DNA damage drugs) in fighting with TNBC.

Additional context to help readers interpret or understand the significance of the work:

This work reported a new mechanism related to TNBC chemo-resistance, which mainly depends on ordered interactions among ITGB4/TNFAIP2/IQGAP1/Rac1 and the following activation of pathways. Thus micro-peptide targeting technique, which is widely used to develop targeted drugs for protein-protein interactions, could show extraordinary potentials and application significance.

At present, cell penetrating peptide, a type of micro-peptide targeting technique, makes functional micro-peptides more stable by cross-linking some amino acid side chains. In recent years, it has been found that binding peptides can not only stabilize peptides, make them easier to enter cells, but also not easy to be hydrolyzed by proteases. At the same time, they have high affinity for targets and can target protein interactions, thus becoming a new way to develop protein interaction targeting inhibitors. To make it easier to enter cells, cell-penetrating peptides can be used in combination, such as HIV TAT. Cell-penetrating peptides can carry a variety of biologically active substances into the cell, is a good targeting drug carrier, with low toxicity, not limited by cell type, into the cell speed and high transduction efficiency. Based on the mechanism reported here, researchers can explore new micro-peptides targeting the interactions between ITGB4 and TNFAIP2 or TNFAIP2 and IQGAP1 to enhance the sensitivity of TNBC cells to drugs by cell-penetrating peptide technology.

---

## [Author Response]

The following is the authors’ response to the original reviews.

**Reviewer #1:**
In this study, Fang H et al. describe a potential pathway, ITGB4-TNFAIP2-IQGAP1-Rac1, that may involve in the drug resistance in triple negative breast cancer (TNBC). Mechanistically, it was demonstrated that TNFAIP2 bind with IQGAP1 and ITGB4 activating Rac1 and the following drug resistance. The present study focused on breast cancer cell lines with supporting data from mouse model and patient breast cancer tissues. The study is interesting. The experiments were well controlled and carefully carried out. The conclusion is supported by strong evidence provided in the manuscript. The authors may want to discuss the link between ITGB4 and Rac1, between IQGAP1 and Rac1, and between TNFAIP2 and Rac1 as compared with the current results obtained. This is important considering some recent publications in this area (Cancer Sci 2021, J Biol Chem 2008, Cancer Res 2023). In addition, some key points need to be addressed in order to support their conclusion in full.

Thanks for your positive comments.

1. It is rarely found studies using the term of "DNA damage drug resistance". Do the authors mean "DNA damage and drug resistance" or "DNA damage-related drug resistance" or "DNA damage-induced drug resistance"? It is better to define "DNA damage drug resistance" in the manuscript if it is not a common term in the field.

We agree with you that the description "DNA damage-related drug resistance" is better so that we revised it uniformly in the manuscript.

2. For Figure 4A, it is stated the IQGAP1 is identified via IP-MS. However, the MS results are not presented in the Figure or in the supplementary. In Figure 4A, only the IP results with silver staining was presented. Moreover, based on the silver staining here, a bunch of proteins were increased in TNFAIP2 overexpression group compared to the vector group. Especially, there is a much clearer band at 52kDa. The authors didn't explain why they chose IQGAP1 and ITGB4 which are less clear than the protein(s) at 52kDa.

Supplementary file 1 is our mass spectrometry results. There are two reasons for choosing ITGB4 and IQGAP1. Firstly, we selected the proteins that indeed interact with TNFAIP2 according to our verification experiments. Secondly, we were interested in the mechanism by which TNFAIP2 promoting DNA damage-related drug resistance, and we found that ITGB4 promoted drug resistance, while IQGAP1 activated Rac1.

3. According to the images in Figure 4C, the efficiency of si-IQGAP1 is limited. The authors could analyze the WB image to confirm the inhibition efficiency of si-IQGAP1.

We analyzed the WB images and the quantitative results are as follows in Author response image 1. The knockdown efficiency is acceptable.

Author response image 1.

**Author response image 1. sa2fig1:** Comparing fitting the experimental molecular displacements with a simple Brownian diffusion model (Figure 3—figure supplement 1C1) against fitting with the optimized HMM two-state diffusion model. Best fit of the same experimental displacement distribution in condensed phase with a two-state HMM model. Red Curve is the HMM model fit obtained by non-linear least squares method using MATLAB. R^2^ = 0.98, RMSE = 184.9..

4. In Figure 5B, I wonder whether the authors can explain why the IgG could immunoprecipitate similar amount of ITGB4 protein as input group.

In this experiment, the Input group had relatively less loading amount (5%), while the IgG group had nonspecific binding.

5. According to the results from Figure 6B, the inhibition efficiency of shITGB4#1 is much higher than shITGB4#2. However, the effects of shITGB4#1 on GTP-Rac1 are similar to or even weaker than those of shITGB4#2 in both HCC1806 and HCC1937. Can this be explained?

The possible reason is that downregulation of ITGB4 expression to a certain level is sufficient to inhibit the activation of Rac1.

6. In Figure 6F, there are double bands for ITGB4 while only one band shows in other Figures. Please find a better representative image here.

ITGB4 has a cleaved band in addition to the main band. These two bands could be separated when we used a low concentration SDS-PAGE gel.

7. In the manuscript, GAPDH, b-Actin and Tubulin are used in different experiments as internal controls. Is there any specific reason to using different internal controls for different experiments here?

There is no specific reason using different internal controls. These experiments were conducted by different person. Each individual chose different internal controls based on the protein sizes.

8. I cannot find Table 1 for the correlation results for TNFAIP2 and ITGB4. I wonder whether Figure 8E is the Table 1 as is mentioned, since it is stated in line 561 that Figure 8E is "the work model of this paper" but actually Figure 8F is. If Figure 8E is the correlation results, I highly recommended the scatter plots graph is used here to present more clear and visualized correlation between TNFAIP2 and ITGB4.

Figure 8E is indeed the correlation result. In addition, Figure 8E could not be presented as scatter plot graph because the pattern of TNFAIP2 and ITGB4 expression is negative or positive according to the determination of IHC results which was carried out by professional pathologists.

9. Throughout the whole manuscript, no description of N number was found in figure legends or in Methods for in vitro experiments. N number is important for statistical analysis.

All our experiments have set up three replicates. We provide this information in figure legends.

**Reviewer #2:**
Breast cancer is the most common malignant tumor in women. One of subtypes in breast cancer is so called triple-negative breast cancer (TNBC), which represents the most difficult subtype to treat and cure in the clinic. Chemotherapy drugs including epirubicin and cisplatin are widely used for TNBC treatment. However, drug resistance remains as a challenge in the clinic. The authors uncovered a molecular pathway involved in chemotherapy drug resistance, and molecular players in this pathway represent as potential drug targets to overcome drug resistance. The experiments are well designed and the conclusions drawn mostly were supported by the data. The findings have potential to be translated into the clinic.

Thanks for your positive comments.

1. In Introduction, the statement of "Breast cancer is the most common malignant tumor in women, and the morbidity and mortality rates of female malignant tumors are ranked first in the world" is inaccurate.

We have revised the description as“Breast cancer is the most commonly diagnosed cancer and the leading cause of cancer death in women”.

2. In Materials and Methods, "Immunopurification and silver staining" is not correct, which should be replaced with "Immunoprecipitation and silver staining".

We replaced the description in the manuscript according to your suggestion.

3. It is unclear Why the authors chose the two TNBC cell lines, HCC1806 and HCC1937, for cell models in this work.

We chose these two cell lines according to our previous work“KLF5 promotes breast cancer proliferation, migration and invasion in part by upregulating the transcription of TNFAIP2” (doi: 10.1038/onc.2015.263. Epub 2015 Jul 20).

41. To demonstrate TNFAIP2 and ITGB4 confer TNBC drug resistance in vivo, the knockdown efficiency of animal experiments was not shown.

The knockdown efficiency of animal experiments was shown below. We added this result into Figure 2-figure supplement 2G and Figure 5-figure supplement 2N.

5. I would strongly suggest the authors seek help from a language editing service to improve the manuscript.

We improved the manuscript by using a professional English language editing service and we have carefully revised the manuscript.

**Reviewer #3:**
In this manuscript, Fang and colleagues found that IQGAP1 interacts with TNFAIP2, which activates Rac1 to promote drug resistance in TNBC. Furthermore, they found that ITGB4 could interact with TNFAIP2 to promote TNBC drug resistance via the TNFAIP2/IQGAP1/Rac1 axis by promoting DNA damage repair.This work has good innovation and high potential clinical significance. However, there are several unsolved concerns that have to be addressed.

Thanks for your positive comments.

1. In the manuscript, there are four drugs used for in vitro cell experiments, why is olaparib (AZD) not used for in vivo animal experiments？

There are two reasons why we did not choose AZD. First，the killing effect of AZD is not as strong as that of BMN. Second, AZD is more expensive than BMN. We finally chose BMN for animal experiments.

2. In Figure 4B, why the immunoprecipitation experiments is done in HCC1806 cell line？

In our previous study “KLF5 promotes breast cancer proliferation, migration and invasion in part by upregulating the transcription of TNFAIP2” (doi: 10.1038/onc.2015.263. Epub 2015 Jul 20), we found that TNFAIP2 knockdown could obviously inhibit the activation of Rac1 in HCC1806 when compared to the result in HCC1937. So, we used HCC1806 cell line to perform the IP-Mass assay.

3. There should be data showing the knockdown effect of TNFAIP2 and ITGB4 in animal experiments.

We addressed the same question above (Reviewer #2, Question#4).

4. When screening the interaction regions between ITGB4 and TNFAIP2, why the TNFAIP2 protein truncation strategy is to delete the N-terminus？

In fact, we also deleted the C-terminus, but the deletion of C-terminus of TNFAIP2 did not affect the interaction.

5. In the manuscript, "input" should be changed to "Input".

We corrected it.

6. There should be a space between "Figure" and numbers.

We add a space between "Figure" and numbers.